# Prediction of Peptide Detectability Based on CapsNet and Convolutional Block Attention Module

**DOI:** 10.3390/ijms222112080

**Published:** 2021-11-08

**Authors:** Minzhe Yu, Yushuai Duan, Zhong Li, Yang Zhang

**Affiliations:** 1Department of Mathematical Sciences, School of Science, Zhejiang Sci-Tech University, Xuelin St., Hangzhou 310018, China; 201930605036@mail.zstu.edu.cn (M.Y.); duanyushuai1022@163.com (Y.D.); 2Institutes of Biomedical Sciences, Fudan University, 138 Yixueyuan Road, Shanghai 200032, China; zhangyang@fudan.edu.cn

**Keywords:** peptide detectability, CapsNet, CBAM, physicochemical properties of residues, amino acid composition, dipeptide composition

## Abstract

According to proteomics technology, as impacted by the complexity of sampling in the experimental process, several problems remain with the reproducibility of mass spectrometry experiments, and the peptide identification and quantitative results continue to be random. Predicting the detectability exhibited by peptides can optimize the mentioned results to be more accurate, so such a prediction is of high research significance. This study builds a novel method to predict the detectability of peptides by complying with the capsule network (CapsNet) and the convolutional block attention module (CBAM). First, the residue conical coordinate (RCC), the amino acid composition (AAC), the dipeptide composition (DPC), and the sequence embedding code (SEC) are extracted as the peptide chain features. Subsequently, these features are divided into the biological feature and sequence feature, and separately inputted into the neural network of CapsNet. Moreover, the attention module CBAM is added to the network to assign weights to channels and spaces, as an attempt to enhance the feature learning and improve the network training effect. To verify the effectiveness of the proposed method, it is compared with some other popular methods. As revealed from the experimentally achieved results, the proposed method outperforms those methods in most performance assessments.

## 1. Introduction

Proteomics is a vital technology in the field of high-throughput experiments. To be specific, protein detection and quantification are vital for gaining more insights into cell biology and human disease [1]. The advancement of mass spectrometry (MS) analysis is of critical significance to provide reliable results at the proteomics level, whereas several problems remain with the existing technology (complex experimental procedures make data processing difficult) (e.g., peptide detectability). The detectability of a peptide is defined as the possibility of observing (or identifying) the peptide from a standard sample mixture. The predicted peptide detectability can be drawn upon to solve shotgun proteomics (e.g., protein inference [2] and label-free quantification [3]). Since there are considerable variables when a certain peptide chain or amino acid sequence is being detected with the mass spectrometer, the characteristics of a set peptide are difficult to quantify [4]. Recently, researchers have proposed a wide range of methods to assess the detectability of peptides by employing standard samples [5] or peptide groups identified in different biological samples [6]. Since there have been numerous positive and negative samples with detectable peptide chains over the past few years, some machine learning (deep learning) methods [7,8] can be exploited to predict the detectability of peptides.

Guruceaga et al. initially defined two types or classes of peptides according to detected and undetected peptides by MS [9]. They extracted features by the physicochemical properties of peptides stored in the AAIndex resource [10], and subsequently used machine learning methods (e.g., SVM and random forest (RF)) for classification. Li et al. [5] proposed a novel algorithm for the iterative learning of peptide detectability from complex samples. Based on their method, 292 features were computed solely from peptide chains and the neighboring residues in proteins as the feature input. Next, a modular neural network was designed to estimate protein quantities and predict the peptide detectability. Zimmer et al. [11] developed an algorithm by complying with the deep fully connected feed-forward neural network, thereby achieving the informed selection of synthetic prototypic peptides to effectively design targeted proteomics quantification assays. They adopted a BioFSharp toolbox to convert a set peptide chain into a feature vector with 45 entries, representing a numerical footprint of physicochemical properties of peptides as the input of the neural network. Wei et al. [12] developed PEPred-Suite, a tool that introduces an adaptive feature representation strategy capable of learning the most representative features exhibited by different peptide types. Thus, it can predict a variety of peptide types simultaneously (e.g., anti-inflammatory peptides and antiviral peptides). Zhang et al. [13] proposed a therapeutic peptide prediction feature by complying with the physicochemical properties of residues, and then adopted the RF prediction method to solve various peptide prediction problems. These researchers finally achieved a satisfactory prediction result. Recently, deep learning methods have been applied for feature extraction and learning and have been extensively employed in biological data prediction and classification [14]. For instance, Guruceaga et al. proposed a deep learning method for the detectability of peptides [15]. They extracted the sequence coding feature and subsequently adopted CNN training to develop a DeepMSPeptide method. By this method, the test dataset in the GPMDB database achieved an accuracy of 79.53%. Cheng et al. [6] proposed PepFormer, a novel type of end-to-end conjoined network that couples with a hybrid architecture of Transformer and gated cyclic units, which is capable of predicting the detectability of peptides by only complying with the peptide chain.

This study introduces a novel integrated learning network framework based on the capsule network (CapsNet) and the convolutional block attention module (CBAM) to predict the detectability of peptides. First, it builds the residue conical coordinate (RCC) feature [16] by complying with the physicochemical properties of residues and combines the statistical information to improve feature extraction. In addition, the amino acid composition (AAC) and dipeptide composition (DPC) and the sequence embedding code (SEC) are fused as the feature input for the detectability prediction of peptides. Next, the mentioned features are split into biological features and sequence features and separately inputted into the neural network model to reduce the influence attributed to the mentioned two types of features. In the proposed neural network framework, it applies CapsNet [17], thereby reducing the impact of losing part of the convolutional and pooling layers of CNN. Furthermore, CBAM [18] is introduced to the capsule network to assign weights to channels and spaces to learn the vital features and optimize the prediction results. The experimentally achieved results verify the effectiveness of this method.

## 2. Results

CapsNet and CBAM are integrated to design a novel neural network model for predicting the peptide detectability. To assess the proposed model, a comparison is drawn for the performance with or without a CBAM module, the experiment is performed using different feature inputs, and different neural network frameworks are compared to verify the effectiveness of the proposed model. Subsequently, this study analyzes the accuracy of the test set with the GPMDB database [3] and compares it with some popular methods. Furthermore, the detectability of different peptides is predicted for additional benchmark tests. In addition, to explore the ability of this model to predict other types of peptides, a test is set to predict whether the predicted peptides are anti-angiogenic peptides or antibacterial peptides.

### 2.1. Comparison with or without CBAM Module and with or without Feature Separation Input

To verify the effect of the CBAM attention module and feature separation input on peptide detectability prediction, three experimental groups are set, i.e., CapsNet + CBAM + feature (biological and sequence feature) separation input, CapsNet + feature separation input, and CapsNet + CBAM + feature combination input. The experimentally achieved results are presented in Figure 1 (Figure 1a draws a comparison of the area under the ROC curve (AUC) and Figure 1b shows a comparison of accuracy (ACC)). When repeating the experiment ten times (the experiment is repeated through the ten-fold cross-validation, and the average and deviations of ten results are taken as the result), box plots are employed to compare AUC and ACC indicators of the mentioned three frameworks. AUC and ACC indicators can reveal the advantage of the attention module of this study and the role of the feature separation input here (AUC and ACC by CBAM module and feature separation input achieve the highest values of 0.862 and 0.801, respectively). This study finds that the feature area receives more attention after convolution with a large degree of discrimination because of the CBAM module, thereby improving the accuracy of the detectability prediction of peptides. In addition, the feature separation operation is exploited to separate the feature of embedding from the features by other biological calculations to reduce the noise influence between two types of features during convolution, which can improve the prediction results as well.

### 2.2. Comparison with Different Input Features

Besides the CBAM module, various input features play different roles in the proposed prediction model. The ablation study is conducted by setting different feature inputs in the proposed model. To be specific, the predicting performance is compared by inputting the residue conical coordinate (RCC) feature, amino acid composition (AAC) and dipeptide composition (DPC), and the sequence embedding code (SEC), respectively. The RCC feature covers the effect of physicochemical properties exhibited by the respective amino acid. The AAC and DPC reflect the effect of the frequency of the respective amino acid and dipeptide on the detectability of peptides. The SEC contains the sequence information of residues in the peptide chain. The comparison results are shown in Table 1, and their ROC curves (the 10-fold cross-validation result is used to calculate the average value of true positive rate and false positive rate to plot the ROC curve) are shown in Figure 2a–c. In Table 1, the mean and standard deviations of ten-fold cross-validation are provided for all different feature inputs. According to the table, the addition of RCC and SEC features optimizes the results of peptide detectability prediction. The addition of AAC and DPC (frequency features) slightly improves the AUC result, whereas it improves the F-score result, which is shown in Figure 2d. Figure 2e provides the precision–recall (PR) curves between the proposed model and other models with different feature input combinations. The accuracy and recall rate of each model are obtained through the average of 10-fold cross-validation. The figure shows that the area under the PR curve by our method is larger than the areas under the PR curves by other models with different feature inputs, namely, the performance of the proposed method is better than that of other models. In the ten-fold cross-validation, the statistically significant difference is analyzed between the proposed model and other models with different feature input combinations. The respective *p*-value in hypothesis testing is less than 0.05, indicating that the proposed model is significantly different compared to the other three models.

### 2.3. Different Prediction Method Comparison for the GPMDB Dataset Test

The proposed model of integrating CapsNet and CBAM is also compared with other popular prediction methods for the detectability of peptides. The dataset is set up by referencing the 1D-2C-CNN method [15], and the 10,000 tryptic peptides are divided into 75% training set and 25% test set. During the training, the hyperparameters of this study are kept to be identical to those in the previous 10-fold cross-validation training model. Table 2 lists the comparison between the experimentally achieved results and other methods (e.g., the method of 1D-2C-CNN by Serrano et al. [15], RF, SvmR, C5, Pls, and Glm methods by Guuruceaga et al. [9], the DNN method by Zimmer et al. [11], and the Gaussian method by Mallick et al. [16]). For the proposed method, the mean and standard deviations of 10-fold cross-validation are calculated as the result. For other methods, their trained model is adopted to acquire the prediction result. It is indicated that our AUC is 1.22% higher than that of the previous best model 1D-2C-CNN, and the other three indicators are better than other models (specificity slightly lower than 1D-2C-CNN). In addition, the statistically significant difference analysis is conducted between the proposed method and 1D-2C-CNN. In other words, the *p*-values of ten-fold cross-validations are determined between the proposed model and 1D-2C-CNN method. The results are presented in Figure 3, which suggest that the proposed model is significantly different from the 1D-2C-CNN method (*p*-value < 0.05). We also conduct a PR curve analysis between the proposed method and 1D-2C-CNN. The precision and recall of each model are obtained by averaging through the 10-fold cross-validation. The result is shown in Figure 4. The PR curve by the proposed method mostly covers the curve by 1D-2C-CNN, which indicates that the performance of the proposed model is better than that of 1D-2C-CNN.

### 2.4. Additional Benchmarks for Testing

The MS evidence of the human proteome offered by the HPP project is applied, and data are extracted from the neXtProt database [19] to build an additional benchmarking. The peptides of this database (also covered in GPMDB, but not related to the training and test sets of peptides according to Ref [15]) fall into three non-overlapping groups: (1) Proteins with MS evidence (PE1) in the existing version (2019-01-11); (2) PE1 proteins in the current release without MS evidence at the beginning of the HPP project (2011-08-23); (3) Missing Proteins (MPs) in the existing version. A total of 8000 peptide chains are selected from the respective group as three benchmark test sets.

Figure 5a,b show the detectability prediction results of peptides with different protein evidence from the three benchmark test sets. For the comparison, the proposed method on the first two datasets PE1 and Detected MPs shows the high detectability, which is consistent with the prediction performance of 1D-2C-CNN on the whole. However, the average probability values predicted by the proposed method on the PE1 and Detected MPs datasets are 0.7143 and 0.838, respectively, which are higher than those predicted by 1D-2C-CNN (0.7039 and 0.7052, respectively). For the Current MPs dataset, since the peptides in this dataset are considered difficult to detect, the lower the detectability value, the better the prediction method will be. As revealed from the comparison result, the proposed prediction model is better than the 1D-2C-CNN method (the proposed method achieves an average probability value of 0.3058 on the Current MPs dataset, while the prediction probability by 1D-2C-CNN is 0.3288).

### 2.5. Additional Datasets for Testing

For the detectability prediction of peptides, it aims to examine the intrinsic characteristics in the peptide chain to distinguish the detectability of peptides. Lastly, an additional two datasets are applied to test the ability of the proposed neural network for searching the mentioned features in the peptide chain. Two datasets include anti-angiogenic peptides (AAP) and antibacterial peptides (ABP). We also compared the proposed method with some popular methods (e.g., PEPred-Suite [12], AntiAngioPred [20], and AntiBP [21]), as shown in Figure 6. It is shown that the proposed method is better than the mentioned latest methods in the AUC indicator: the AUC by the proposed method reached 0.811 on the AAP dataset (PEPred-Suited and AntiAngioPred reached 0.804 and 0.742, respectively), and 0.979 on ABP (PEPred-Suite and AntiBP reached the same as 0.976). This verifies that our network model can also be applied for predicting other therapeutic peptides.

## 3. Discussion

This study proposes a neural network model integrated with CapsNet and CBAM to predict the detectability of peptides. It constructs the residue conical coordinate (RCC) feature, amino acid composition (ACC) and dipeptide composition (DPC), and sequence embedding code (SEC) to generate peptide chain features for the proposed network. For the mentioned features, they fall to biological feature and sequence feature by separate inputting to reduce the influence among the mentioned features. When using the CapsNet network, it reduces the impact of data loss in the pooling layer after convolution. In addition, a CBAM attention module is added to assign weights to channels and spaces to learn important features. The proposed model is compared with other extensively used deep learning frameworks (e.g., 1D-2C-CNN [15], RF, SvmR, DNN, C5, Pls, Glm, and Gaussian [9,11,16]). It is suggested to outperform other methods in main assessment indicators (AUC, accuracy, specificity, sensitivity, and F-score), which verifies the effectiveness of the proposed method. Consequently, it can act as a valid supplementary method for peptide detectability prediction and be applied in proteomics and other related fields.

Since the length of the peptide chain is short and its length is different, the proposed method selects a truncated fixed length for calculation. It will be our future work to find a feature extraction method that is more suitable for characteristics with different peptide chain lengths. In addition, this method selects two biological features and directly joins them together. We need to extract other biological features and explore a valid feature fusion method for peptide detectability prediction so the mentioned biological features can be more effectively integrated for training. With the growing popularity of neural networks, we also look for a more suitable network model for the classification of peptide chains. In addition, the CapsNet + CBAM model is an end-to-end model. It is expected that this model can be employed to predict other peptide chains (e.g., cell-penetrating peptides).

## 4. Materials and Methods

### 4.1. Dataset

The dataset applied for training and testing in this study originates from the GPMDB database [3], involving mass spectrometry data and detection frequencies for proteins identified by mass spectrometry. The method of generating the dataset was proposed by Guruceaga et al. [9], capable of classifying trypsin-digested peptides according to the number of peptides observed in proteomics experiments and comparing the characteristics exhibited by the most observed peptides and the less observed peptides. A total of 100,000 tryptic peptides are selected as the dataset, and then the ten-fold cross-validation is performed for the experiment (i.e., it is divided into 10 parts, taking turns using 9 parts as the training data and 1 part as the test data for repeating the experiment). The respective test will obtain the corresponding prediction result. The average of the prediction results by 10 times is used as the final prediction result.

### 4.2. Feature Selection

Feature selection refers to a vital step in the deep learning of a neural network framework. The property of amino acids and sequence information are exploited to generate three features, i.e., the residue conical coordinate (RCC) feature, amino acid composition (AAC) and dipeptide composition (DPC), as well as the sequence embedding code (SEC) feature.

#### 4.2.1. RCC Feature Based on Physicochemical Properties of Amino Acids

The respective amino acid has specific physicochemical properties, thereby affecting the characteristics of the peptide and being critical in determining the structure and function of the peptide. The 100 physicochemical properties of amino acids are drawn upon for feature extraction. Due to the significant numerical difference between the physicochemical properties, the standardized operation is exploited to process the data and improve the comparability of data. Furthermore, since some amino acids exhibit consistent characteristics, the mentioned amino acids are classified into 4 types [17] (Table 3).

Zhang et al. [16] proposed a residue conical coordinate (RCC) feature by mapping the respective amino acid to a point in a three-dimensional (3D) space. Subsequently, they integrated this spatial coordinate feature with the RF method to conduct a satisfying prediction for carbonylation site identification.

To be specific, to obtain the vital features of proteins with the use of a simple and effective mapping method, the following two hypotheses were proposed. (1). The amino acids in the same group are distributed on an identical conical surface since they exhibit consistent characteristics. The amino acids in the same group are distributed on the same conical surface because they show similar characteristics. For instance, as eight amino acids (A, V, L, I, P, F, W, and M) pertain to Class I, they are mapped on the identical cone surface ($\varphi =\varphi_{1}$) (Figure 7). Likewise, the other three types of amino acids are mapped on their respective cones. (2). To reveal the difference between amino acids, the radius vector length *r* of the conical surface is set in accordance with the molecular weight of the corresponding amino acid (http://lin-group.cn/server/iCarPS/download.html (accessed on 4 November 2021)). Following such a mapping rule, the similarity of similar amino acids and the difference between different amino acids can be indicated simply and effectively.

Therefore, by combining the physicochemical properties and the above mapping rules, each amino acid can be converted into a three-dimensional vector. The formula is as follows:(1)$$\{\begin{array}{c} x_{ij}=r_{ij}\times \sin\varphi_{i}\times \cos\theta_{ij}\\ y_{ij}=r_{ij}\times \sin\varphi_{i}\times \sin\theta_{ij}\\ z_{ij}=r_{ij}\times \cos\varphi_{i} \end{array}\;\;\;\varphi_{i}\;\in \left[0,\pi \right]\;,\;\theta_{ij}\in \left[0,2\pi \right]$$
where $r_{ij}$ expresses the molecular weight of the *j*-th residue $\left(j=1,2,\cdots ,L_{i}\right)$ in the *i*-th category (i=1,2,3,4) of the amino acid classification; $L_{i}$ denotes the number of amino acids in the *i*-th group; and $\varphi_{i},\theta_{ij}$ are defined below:(2)$$\varphi_{i}=\pi \times \left|\sin\frac{{\bar{d}}_{i}}{\left(\frac{1}{4}\sum_{1}^{4}{\bar{d}}_{i}\right)\times \sqrt{\frac{1}{4}\sum_{1}^{4}{\left({\bar{d}}_{i}-\frac{1}{4}\sum_{1}^{4}{\bar{d}}_{i}\right)}^{2}}}\right|$$
(3)$$\theta_{ij}=\pi +2\times \tan^{-1}\;\frac{\sum_{m=1}^{9}PC_{jm}-{\bar{d}}_{i}}{\sqrt{\frac{1}{L_{i}}\sum_{j=1}^{L_{i}}{\left|\sum_{m=1}^{9}PC_{jm}-{\bar{d}}_{i}\right|}^{2}}}$$
where ${\bar{d}}_{i}=\frac{1}{L_{i}}\overset{L_{i}}{\underset{j=1}{\sum }}\overset{9}{\underset{m=1}{\sum }}PC_{jm}$; $PC_{jm}$ represents the *m*-th (m=1,2,⋯,9) physicochemical properties of the *j*-th $\left(j=1,2,\cdots ,L_{i}\right)$ residue in the *i*-th category of amino acid classification. In the mentioned conversion process, since only 9 physicochemical properties of amino acids are applied for the calculation, the yielded feature matrix is relatively small for the proposed deep learning network model. This study improves the construction of the RCC feature that fully extracts the physicochemical properties of the residue. In other words, 100 physicochemical properties fall into 10 groups, each of which covers 10 physicochemical properties [13]. Subsequently, 10 sets of residue conical coordinate features are yielded by Equations (1)–(3). Such a measure is capable of increasing the feature dimension generated by the peptide chain and ensuring the feature proportion in the proposed neural network training. Furthermore, we combine statistical information (e.g., variance, skewness, and kurtosis) to extract features exhibited by peptide chains.

To be specific, the *L*-length peptide chain can be transformed as a 1 × 3*L* matrix (e.g., $P={\left[x_{1},y_{1},z_{1},\cdots ,x_{L},y_{L},z_{L}\right]}^{T}$). The overall geometric center of the sequence $\bar{u}\left(\bar{x},\bar{y,}\bar{z}\right)$ is first determined as:(4)$$\bar{u}=\frac{1}{L}\overset{L}{\underset{k=1}{\sum }}u_{k}\;$$
where$\;u_{k}$ represents the *k*-th 3D coordinate point $\left(u_{k}=x_{k},y_{k},z_{k}\right)$.

Subsequently, the residues in the peptide chain are classified into 4 types (Table 3), and the geometric center ${\bar{u}}_{i}$ of the *i*-th category is determined (i=1,2,3,4):(5)$${\bar{u}}_{i}=\frac{1}{v}\sum_{n=1}^{v}u_{in}\;$$
where $u_{in}$ denotes the coordinates of the *n*-th residue pertaining to the *i*-th category of amino acid classification; *v* represents the total number of the *i*-th residue category.

Lastly, the overall geometric center of the peptide chain and the geometric center of the amino acid classification are adopted to determine the statistical characteristics of variance, skewness, and kurtosis [13] as:(6)$$\sigma^{2}=\frac{1}{4}\sum_{i=1}^{4}{\left({\bar{u}}_{i}-\bar{u}\right)}^{2}$$
(7)$$g=\frac{\frac{1}{4}\sum_{i=1}^{4}{\left({\bar{u}}_{i}-\bar{u}\right)}^{3}}{\sigma^{3}}$$
(8)$$h=\frac{\frac{1}{4}\sum_{i=1}^{4}{\left({\bar{u}}_{i}-\bar{u}\right)}^{4}}{\sigma^{2}}$$

For each peptide chain, Equation (5) is used to generate a 1 × 12 dimensional vector according to the amino acid classification (4 types), and Equation (4) and Equations (6)–(8) are adopted to generate a 1 × 12 dimensional vector, so the total combination is a 1 × 24 dimensional vector. Since there are 10 groups of physicochemical properties exhibited by amino acids, a 1 × 240 dimensional feature vector is lastly obtained.

#### 4.2.2. Amino Acid Composition and Dipeptide Composition

Amino acid composition (AAC) acts as a vital feature of a single amino acid, and dipeptide composition (DPC) represents the proportion information of adjacent amino acids in the peptide chain. Both of the mentioned features can indicate the property of the entire peptide chain [22,23]. AAC denotes the percentage of a single amino acid in a set peptide chain and can be calculated by:(9)$$\mathrm{AAC}\left(i\right)=\frac{\mathrm{Frequency}\;\mathrm{of}\;\mathrm{amino}\;\mathrm{acid}\left(i\right)}{\mathrm{Length}\;\mathrm{of}\;\mathrm{the}\;\mathrm{peptide}}\;\;\;i=\left(1,2\cdots 20\right)$$
where i represents the respective amino acid, and the length of the AAC generation vector is 1×20.

Because AAC does not consider the order of amino acids in the peptide chain, the DPC feature is added in our feature information. DPC denotes the probability of two adjacent amino acids appearing in the entire peptide chain, which has a fixed length of 400 features, as expressed below:(10)$$\mathrm{DPC}\left(j\right)=\frac{\mathrm{Total}\;\mathrm{number}\;\mathrm{of}\;\mathrm{dipeptides}\left(j\right)}{\mathrm{Total}\;\mathrm{number}\;\mathrm{of}\;\mathrm{all}\;\mathrm{possible}\;\mathrm{dipeptides}}$$
where j represents one of 400 dipeptide compositions.

#### 4.2.3. Neural Network Embedding

For biological sequences, the one-hot encoding is the most used feature as input in the neural networks [24]. When processing the peptide chain, the one-hot encoding turns out to be a 20-dimensional vector composed of 0 and 1, which is sparse and fails to show the neighboring relationship of the respective residue. Here, the sequence embedding code (SEC) [15] is adopted to calculate the neighboring relationship of the residues and control the dimensionality of the vector. SEC acts as a method to convert discrete variables into a continuous vector representation. Since the SEC is learnable, the representations of more similar residues will be closer to each other in the embedding space during the continuous training process. For this reason, the SEC is selected as the feature input to participate in training in the neural network, and the respective amino acid embedding is set as a 20-dimensional vector.

### 4.3. Neural Network Architecture

Convolutional Neural Network (CNN) refers to an extensively applied network framework. Its capacity of solving classification is achieved primarily by using the convolutional and pooling layers to reduce the spatial size of data passing through the network. On that basis, the field of view of the neurons in the neural network increases, and the high-level features of the input area are detected. However, there exist two main problems in such a process. First, the convolution is locally connected and parameter-sharing, while it does not consider the correlation and mutual positional relationship between different features. Second, it keeps only the most active neurons in the process of maximum pooling and pushes them to the next layer, thereby causing the loss of valuable spatial information.

This study uses the capsule neural network (CapsNet) [17] to solve the mentioned problems. The difference between this neural network and the general neural network is that its neuron is a vector (a set of values) (vector neuron) rather than a scalar (single value). The vector is capable of representing a wide range of characteristics of a protein, and the modulus length of the vector can be exploited to measure the probability of the respective category. The greater the modulus value, the greater the probability of pertaining to a set category will be. Compared with CNN neurons with scalar input and output, the input and output vectors of the capsule network cover more feature space information. Furthermore, the capsule network substitutes the maximum pooling with a dynamic routing mechanism. Compared with the popular maximum pooling, it can retain the weighted sum of the features of the previous layer, whereas the maximum pooling only retains the most active neurons.

The proposed neural network model comprises a CNN convolutional layer, a CBAM module, a convolutional capsule layer (PrimaryCaps), as well as a fully connected capsule layer (BindCaps) (Figure 8). First, the RCC, AAC, and DPC features of the peptide chain are employed as the biological features, and the SEC feature is inputted into the CNN layer as the sequence feature. The mentioned features are convolved via CNN to extract local feature information and then inputted to the CBAM module. CBAM combines the channel attention mechanism and the spatial attention mechanism [18]. In the CBAM module, the local feature information will be multiplied by the different weights trained, which can arouse the attention to vital feature information and suppress the unimportant feature information. The CBAM module is presented in Section 2.4. Moreover, the detailed feature transfer in the neural network is expressed in Section 4.5.

Subsequently, the biological and sequence features processed by CBAM are converted into 32 × 16 × 544, 32 × 16 × 704 feature maps and then inputted to the PrimaryCaps layer. To be specific, 32 represents the number of peptide chains input in one training, 16 indicates the number of channels in the PrimaryCaps layer, and 544 and 704, respectively, represent the size of the biological and sequence feature information obtained after CBAM. For the feature map corresponding to the respective peptide chain, the calculation of the iterative dynamic routing between the PrimaryCaps and BindCaps layers is illustrated in Figure 9. To be specific, the PrimaryCaps layer comprises 1248 (544 + 704) capsules $u_{i}$ (each $u_{i}$ denotes a 16-dimensional vector), and each capsule is multiplied by the weight matrix $W_{I,j}$ (the size 16 × 32, *i* = 1,…,1248, *j* = 1,2) to obtain ${\hat{u}}_{i|j}$. ${\hat{u}}_{i|j}$ is multiplied by the weighted sum of the parameters $c_{i,j}$ to yield $s_{j}$. Subsequently, through the nonlinear “Squeeze” function, the capsule vector in BindCaps is obtained after a set number of iterations (epoch). To be specific, the “squeeze” function [15] is adopted to scale the length of the output vector of each capsule to between [0, 1] and keep the direction unchanged, as expressed below:(11)$$v_{j}=\frac{||s_{j}||{}^{2}}{1+||s_{j}||}\cdot \frac{s_{j}}{\left||s_{j}|\right|}$$
where $s_{j}\;$and $v_{j}$, respectively, represent the input and output of the *j*-th capsule.

Lastly, the two output capsules calculated by using the routing algorithm are integrated to obtain two 32-dimensional BindCaps, corresponding to two possible markers of the input protein sequence, i.e., peptide detectability and undetectability. The modulus of the 32-dimensional vector represents the detectability and undetectability probability of the peptide. To be specific, the L2-norm of two 32-dimensional vectors is calculated, and the detectability of the peptide is predicted by comparing the magnitude of the two probability values.

In the proposed neural network framework, the loss function adopted to train the CapsNet model is the sum of two separate losses in BindCaps. For each BindCaps, the separate loss $L_{k}$ is expressed as:(12)$$L_{k}=T_{k}\max{\left(0,0.9-||v_{k}||\right)}^{2}+0.5\left(1-T_{k}\right)\max{\left(0,||v_{k}||-0.1\right)}^{2}$$
where $v_{k}$ denotes the vector output by BindCaps. If the peptide is detectable, $T_{k}=1$; otherwise, it is 0.

### 4.4. CBAM Module

In the proposed neural network, to extract vital features distinguishing the detectability of peptides, the CBAM module is added after the first layer of convolution. CBAM refers to an attention mechanism module combining space and channel information. The spatial module assigns different weights to the identical dimension of biological features and sequence features, while the channel attention assigns different weights to the feature map channels after convolution, which is inconsistent with SENet [25] that only focuses on the channel attention module. CBAM focuses on both space and channel information, so it can more effectively predict the detectability of peptides. The specific process of this module is that the first layer of CNN processes the feature map $F\in R^{C\times H\times W}$ as the input. Subsequently, the channel attention module $M_{C}\left(F\right)$ and the spatial attention module $M_{S}\left(F\right)$ of CBAM are employed to set different weights to the feature map (Figure 10). The complete process can be written as:(13)$$F^{\prime }=M_{C}\left(F\right)\times F\;\;F^{\prime \prime }=M_{S}\left(F^{\prime }\right)\times F^{\prime }$$
where × denotes the element-wise multiplication.

According to the CBAM module, the channel attention module is adopted to compress the feature map of the peptide in the spatial dimension to yield a one-dimensional vector. Next, the vector is multiplied with the feature map of the initial input peptide element by element (Figure 11). During channel compressing in the spatial dimension, two pooling operations are considered, i.e., the average pooling $F_{avg}^{c}$ and the maximum pooling $F_{max}^{c}$. They can be exploited to aggregate the spatial information of the feature-mapped peptides, and then the information is transmitted to a shared fully connected network. The size of the hidden layer is set to $R^{\frac{C}{r}\times 1\times 1}$, where *r* denotes the reduction rate. The calculation formula is:(14)$$M_{C}\left(F\right)=\sigma \left(\mathrm{MLP}\left(\mathrm{AvgPool}\left(F\right)\right)+\mathrm{MLP}\left(\mathrm{MaxPool}\left(F\right)\right)\right)$$
where σ denotes the sigmoid function and *F* represents the peptide feature after CNN processing. For the feature map after convolution, the channel places its stress on which channel is critical to the detectability of peptides, and average pooling has feedback for each pixel on the feature map. In addition, maximum pooling is applied for the gradient backpropagation calculation, and only the place with the largest response in the feature map achieves the gradient feedback.

The spatial attention module is exploited to compress the channel (Figure 12). Two pooling (average pooling and maximum pooling) operations are used to aggregate the channel information of the feature map and then merged to yield a 2-channel feature map. After the convolutional layer is passed through and activation function (sigmoid function) operation is achieved, the weight coefficient is obtained and then multiplied with the first input feature to obtain the final output feature. The calculation formula is written below:(15)$$M_{S}\left(F\right)=\sigma \left(f^{7\times 7}\left(\left[\mathrm{AvgPool}\left(F\right);\mathrm{MaxPool}\left(F\right)\right]\right)\right)$$
where σ denotes the sigmoid function, $f^{7\times 7}$ represents the convolution operation, and the filter size is 7×7. For spatial attention, the focus is placed on which part of the corresponding convolution feature map of a peptide is critical to predicting the detectability of the peptide. Up-regulating the weight of the more important areas of the convolution feature map can optimize the detectability exhibited by the peptide.

### 4.5. Network Parameter Setting and Feature Transfer

In the proposed framework, the corresponding parameter setting and the transfer process of features in the neural network are expressed below. The batch size input for training is set to 32 peptide chains. At the first network layer, the feature size of the biometric input is 32×33×20, the convolution kernel is 9×9 in size, and the resulting feature input is 32×256×27×14. The sequence feature input has the feature size of 32×50×20, the size of the convolution kernel is 7×7, and the resulting feature input is 32×256×42×12. At the second layer of the CBAM attention module, the reduction rate is set to 16, and the activation function is ReLU under the channel attention module, while the size of the first convolution kernel layer is 7×7 according to the spatial attention module. Based on the attention module, the feature input and output dimensions remain constant. At the third layer of PrimaryCaps, it is implemented by using 16 filters with the size of 9 and 7, respectively, and the stride is set to 2 in the respective capsule. The tensor generated by the sequence feature is 32×256×17×2, and the tensor generated by the biological feature is 32×256×11×4. The output feature sequence is changed to 32×16×544, and the biological feature is altered to 32×16×704, where 32 represents the training data, 16 denotes the attribute of each sequence trained by the capsule network, and 544 and 704 express the data trained by different weights. Next, at the fourth layer BindCaps, the features are calculated as two 32-dimensional vectors by using the routing algorithm, corresponding to the two classifications applied for prediction. The Adam optimizer [26] is used for the training, the number of epochs is set to 6, the initial learning rate is set to 0.001, and the loss function is set by complying with Equation (12). The ε hyperparameter of the capsule network is set to $1\times 10^{-8}$, the routing parameter is set to 3, and the reduction parameter of the squeeze-and-excitation (SE) layer in CBAM module is set to 16.

### 4.6. Evaluation Index

The detectability prediction of peptides falls into 4 types, i.e., (1) True Positive (TP): the detectability of the detected peptide, consistent with the detectability of the actual peptide; (2) False Positive (FP): the detectability of the detected peptide, whereas the match set by the undetectability of the actual peptide is incorrect; (3) True Negative (TN): the undetectability of the detected peptide, complying with the undetectability of the actual peptide; (4) False Negative (FN): the undetectability of the detected peptide, as well as the detectability of the actual peptide.

The assessment indicators involved in this study include the area under the ROC curve (AUC), accuracy, specificity, sensitivity, and F-score [27]. To be specific, AUC is defined as the area under the ROC curve. The value of this area will not exceed 1. Since the ROC curve is generally above the line *y* = *x*, the value range of AUC is generally between 0.5 and 1. The AUC value acts as the assessment criterion since in many cases the ROC curve does not clearly indicate which classifier performs better; however, as a value, a classifier with a larger AUC is more effective. Sensitivity: The degree of a positive reaction to a real target. The higher the sensitivity, the easier it will be to identify the target. Specificity: The degree of negative reaction to false targets. The higher the specificity, the less likely there will be false positives, and there are only positive reactions for specific scenarios, i.e., strong screening ability or high pertinence. F-score refers to an index applied in statistics to measure the accuracy of a two-classification model. It considers the accuracy rate and recall rate of the classification model. F-score can be considered a harmonic average of model accuracy and recall that ranges from 0 to 1. The calculation formulas for the above assessment indicators are presented below:(16)$$\mathrm{Accuracy}=\frac{\mathrm{TP}}{\mathrm{TP}+\mathrm{FP}}$$
(17)$$\mathrm{Sensitivity}=\frac{\mathrm{TP}}{\mathrm{TP}+\mathrm{FN}}$$
(18)$$\mathrm{Specificity}=\frac{\mathrm{TN}}{\mathrm{FP}+\mathrm{TN}}$$
(19)$$F-\mathrm{score}=2\cdot \frac{\mathrm{Accuracy}\cdot \mathrm{Recall}}{\mathrm{Accuracy}+\mathrm{Recall}}$$

## Figures and Tables

**Figure 1 ijms-22-12080-f001:**
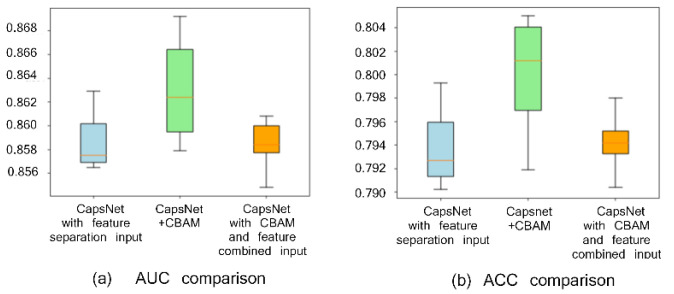
Comparison with or without CBAM and with or without feature separation input on the peptide detectability prediction for the test set from GPMDB. The blue, green, and orange groups are the results by CapsNet + feature separation input, CapsNet + CBAM + feature separation input, and CapsNet + CBAM + feature combined input, respectively.

**Figure 2 ijms-22-12080-f002:**
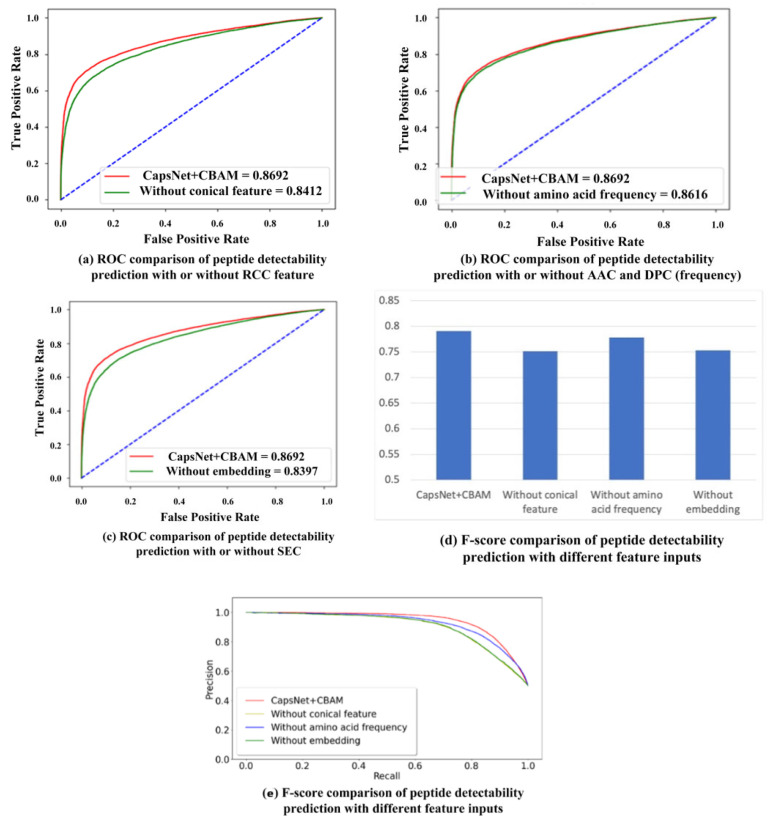
Performance comparison with different feature inputs for peptide detectability prediction on the test set from GPMDB.

**Figure 3 ijms-22-12080-f003:**
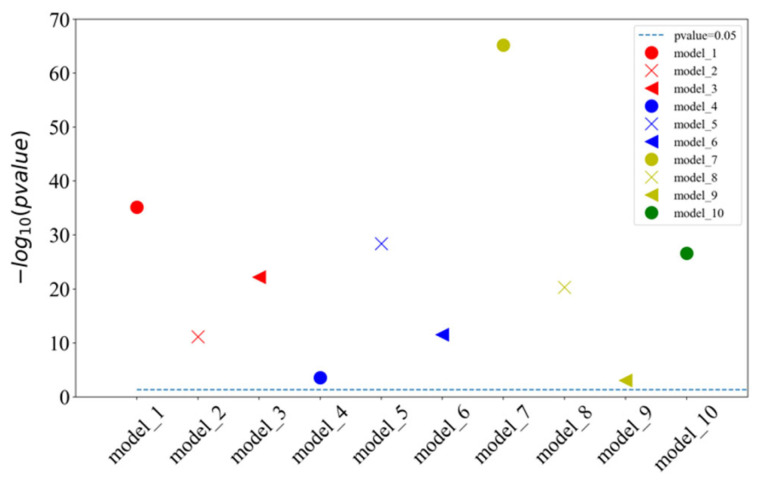
Statistically significant difference analysis of ten-fold cross-validation between our model and 1D-2C-CNN method. Model_1, model_2,…, model_10 are the models of CapsNet + CBAM by the ten-fold cross-validation. When *p*-value is less than 0.05, the difference between two models is significant. Since the *p*-values calculated by ten-fold cross-validation models are quite different, we provide the result of $-log_{10}$ (*p*-value) instead of *p*-value in this figure. When the *p*-value is higher than the baseline (*p*-value = 0.05), it indicates that the ten-fold cross-validation of our model is significantly different from the 1D-2C-CNN method.

**Figure 4 ijms-22-12080-f004:**
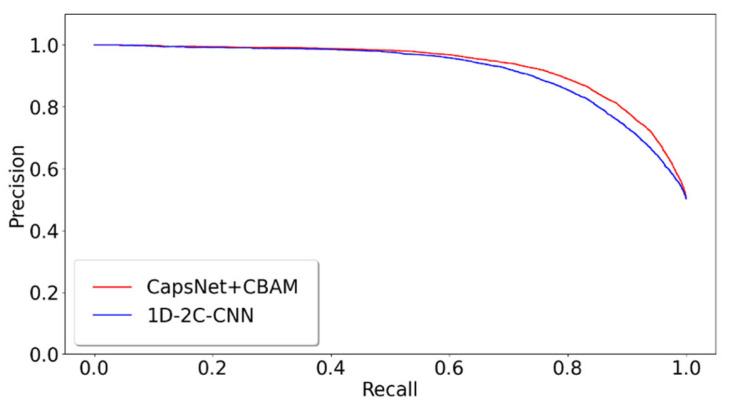
PR comparison of peptide detectability prediction between CapsNet + CBAM and 1D-2C-CNN.

**Figure 5 ijms-22-12080-f005:**
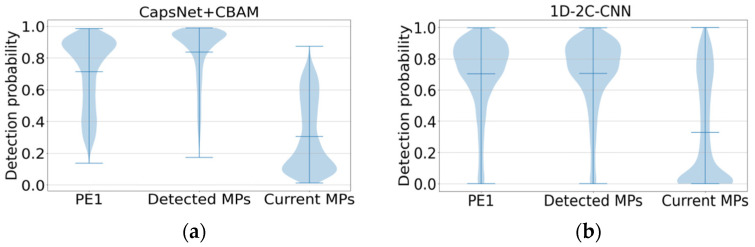
Detectability prediction comparison of peptides with different protein evidence from 3 benchmark test sets. (**a**) Additional benchmark result by our method. (**b**) Additional benchmark result by 1D-2C-CNN.

**Figure 6 ijms-22-12080-f006:**
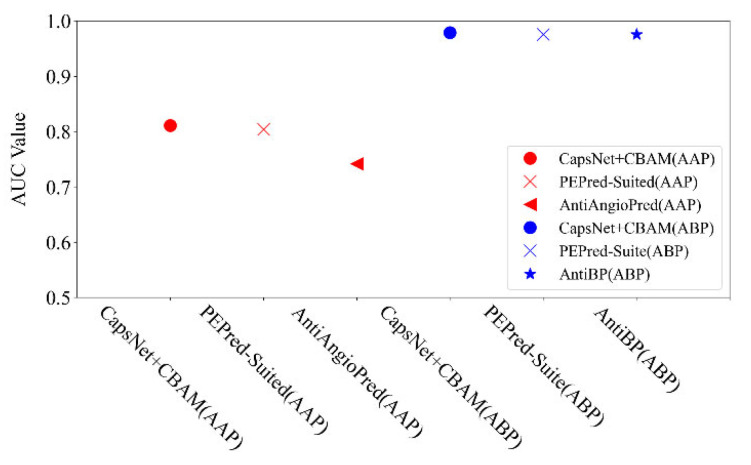
Comparison of AUC indicators by different methods on AAP and ABP datasets.

**Figure 7 ijms-22-12080-f007:**
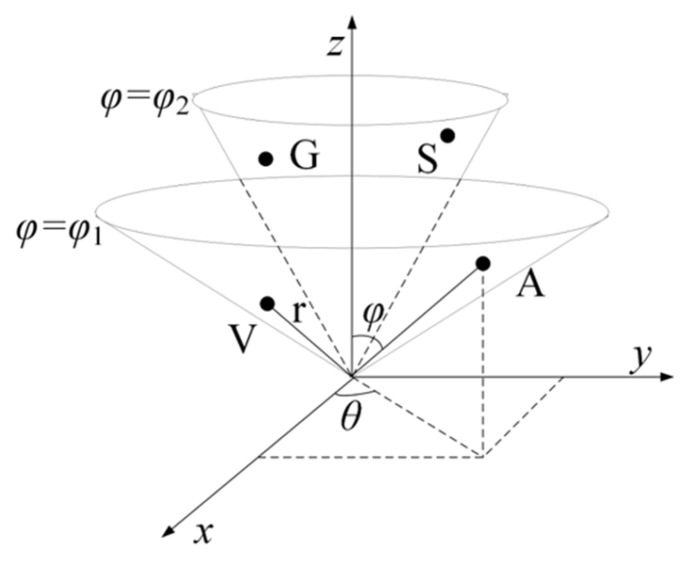
Schematic illustration of the 3-dimensional conical representation for characterizing amino acid residues. A, V, G, and S are the first two amino acids in the Non-polar residue group and Polar residue group. $\varphi_{1}\;$and $\varphi_{2}$ represent the conical surface, which is formed by projection of amino acids in the corresponding group.

**Figure 8 ijms-22-12080-f008:**
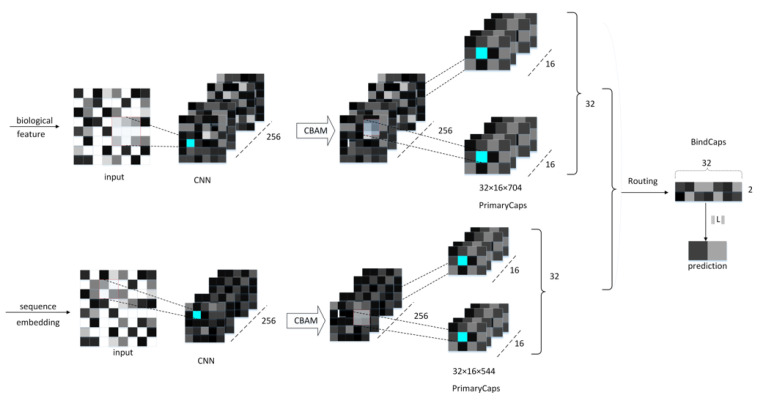
The neural network framework of the proposed method.

**Figure 9 ijms-22-12080-f009:**
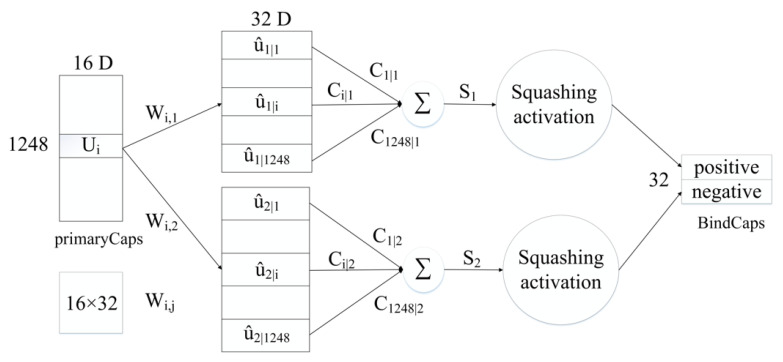
Routing calculation process between PrimaryCaps and BindCaps.

**Figure 10 ijms-22-12080-f010:**
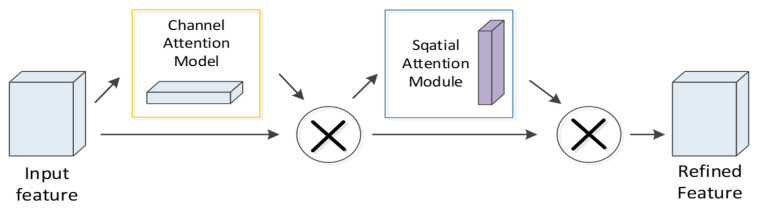
CBAM module.

**Figure 11 ijms-22-12080-f011:**
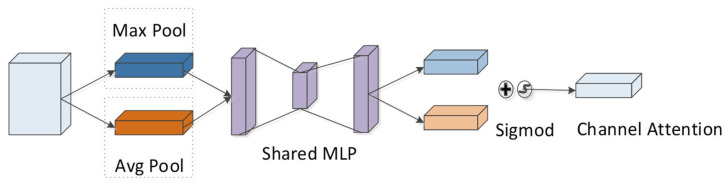
Channel attention module in CBAM.

**Figure 12 ijms-22-12080-f012:**
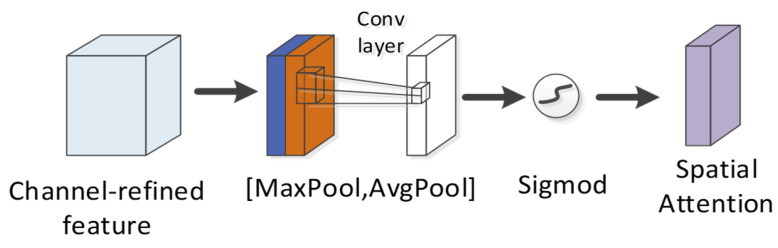
Spatial attention module in CBAM.

**Table 1 ijms-22-12080-t001:** Performance comparison by different feature inputs for peptide detectability prediction on the test set from GPMDB.

Model	AUC	Accuracy	Specificity	Sensitivity	F-Score
**CapsNet + CBAM**	0.8692±0.0029	0.8063±0.0027	0.8819±0.0152	0.7308±0.0140	0.7906±0.0051
**Without con** **ical feature**	0.8412±0.0040	0.7748±0.0040	0.8682±0.0266	0.6814±0.0219	0.7516±0.0090
**Without amino acid frequency**	0.8615±0.0045	0.7978±0.0042	0.8860±0.0225	0.7096±0.0196	0.7782±0.0084
**Without embedding**	0.8396±0.0051	0.7755±0.0053	0.8663±0.0201	0.6847±0.0197	0.7531±0.0072

**Table 2 ijms-22-12080-t002:** Performance comparison by different classifier algorithms for peptide detectability prediction on the test set from GPMDB.

Model	AUC	Accuracy	Specificity	Sensitivity	F-Score
**CapsNet + CBAM**	0.8692±0.0027	0.8050±0.0026	0.8823±0.0147	0.7278±0.0136	0.7887±0.0049
**1D-2C-CNN** [15]	0.8570	0.7953	0.8880	0.7027	0.7744
**RF** [9]	0.7549	0.6924	0.7746	0.6103	0.6649
**SvmR** [9]	0.7384	0.6813	0.7830	0.5797	0.6453
**DNN** [11]	0.7360	0.6692	0.6813	0.6572	0.6659
**C5** [9]	0.7312	0.6644	0.6513	0.6775	0.6687
**Pls** [9]	0.6350	0.6043	0.6396	0.5690	0.5898
**Glm** [9]	0.6349	0.6036	0.6426	0.5646	0.5875
**Gaussian** [16]	0.6342	0.5983	0.6121	0.5845	0.5927

**Table 3 ijms-22-12080-t003:** Classification of 20 amino acids.

Groups	Description	Amino Acids
**Class Ⅰ**	Non-polar residues	A, V, L, I, P, F, W, M
**Class Ⅱ**	Polar residues	G, S, T, C, Y, N, Q
**Class Ⅲ**	Basic residues	K, R, H
**Class Ⅳ**	Acidic residues	D, E

## Data Availability

The datasets and program used in this study are available at https://github.com/yuminzhe/yuminzhe-Prediction-of-peptide-detectability-based-on-CapsNet-and-CBAM-module (accessed on 4 November 2021).

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
