# Peer review of "Prediction of Peptide Detectability Based on CapsNet and Convolutional Block Attention Module"

_ijms, 2021, doi:10.3390/ijms222112080_

Round 1

Reviewer 1 Report

Some of my previous comments have been addressed well. A few points should be addressed more to improve the quality of paper:

1. Web-server should be developed and released to support users test their sequences, especially for biological users without mathematical/programming knowledge.

2. ROC comparison (Fig. 2) did not show many differences between their method and other methods. How to convince that their method is better than the others?

3. Besides ROC curve, the authors should report Precision-Recall curves also.

4. More references on deep learning or CNN-based bioinformatics should be added, i.e., PMID: 31920706, PMID: 33539511.

5. Did the authors plot ROC curves from cross-validation results?

6. In the metrics, F-score is different from F1-score, the authors should make it consistent.

Reviewer 2 Report

The manuscript titled “Prediction of peptide detectability based on CapsNet and CBAM” reports application of novel machine learning algorithms of capsule neural networks and convolutional block attention module to the prediction of peptide detectability (in mass spectrometry analysis) task. This task remains to be of current interest and no high quality predictive systems are developed for today. Capsule network is a novel type of ANN that demonstrated its strength in the recognition of images with multiple objects that are subject to spatial perturbations. This is achieved due to segmentation and pooling processes realized in CapsNet approach. This idea indeed correlates with the inner nature of mass spectrometry, where the data is obtained in the form of masses of small charged fragments and then requires to be pooled to identify a substance or mixture content.

The methods section is written clearly and corresponds to current state-of-art. Common drawback in artificial intelligence studies is the lack of enough details to reproduce the reported computational experiments. However, this study is supplemented with publicly available source code in Jupyter Notebook (Python) for both the data and algorithms used, and final models obtained. Here I want to bring attention that generated pdf used for my review has incorrect (truncated) URL to the source code, though copy and pasting the needed text to browser can overcome that issue. Hope this point will be addressed in the final draft.

The authors obtained fruitful results and showed that the proposed framework leads to higher predictive scores than other popular methods used in literature. The limitation of the study are adequatly presented and ways of further improvement are reported. Few issues in the manuscript need to get an attention.

  • The “CBAM” abbreviation in title can confuse some readers, since “CBAM” is used more often for Concerns based adoption model. The proposal is to write “convolutional block attention module” in the title in the full form.
  • The authors are using label “Ours” in figures across the manuscript. The figures generally should be informative even outside the paper, so this would be confusing to a reader. Please, find the proper way to label models unambiguously (e.g. CapsNet+CBAM).
  • The F-score is used in the manuscript among the main performance indicators. Though F-score is trully common classification performance measure, it has a drawback in not including true negatives into account. In some settings it is justified, for example in anomaly detection or disease diagnostics. However, in the case of detectability prediction the information that a protein is indetectable is even more valuable than it is detectable. The manuscript will be improved if some justification for choosing of F-score will be added, or another, more balanced performance indicator will be used.
  • P-values are used in the manuscript to compare performance in cross-validation runs with other methods. Actually, p-values were invented for very different purpose, and there is no sense to compare different p-values, especially small enough, and taking logarithms is not helpful for that at all. In machine learning studies the model is generated through continuous optimization and selection, it does not come from some predefined hypothesis, and thus criterion like p<0.05 is not applicable at all. It should be accuracy, or AUC, or some other performance indicator that is used to compare models. Figures and discussion that relates to p-values is recommended to be overwritten to abandon p-values utterly.
  • Figure 3 in general is not informative at all: the x axis is used to present different models from 10-fold cross-validation, and there is no information in the number of a model. All this information about cross-validation performance can be presented in the form of median and interval values in text. Once again, p-values should not be used here as well.
  • The lines that connect AUC values of different models in the Figure 5 are redundant. There are no states (points) between different models, which could be represented with values on those lines. The pure dot plot or bar plot is enough for this purpose. The Y axis in the Figure 5 should start at 0.5 which is a baseline value. Starting it with higher values visually overestimates the difference between models.

In general, the manuscript is clearly written and can add value to protein detectability issue. It is also welcomed for correctly applying ideas of CapsNet and CBAM to real-world tasks and popularizing these novel methods. The manuscript can be published after major revision addressing the comments above.

Round 2

Reviewer 1 Report

My previous comments have been addressed well.

Reviewer 2 Report

The authors took into account all the comments. The paper is a solid work and may be accepted.

Round 3

Reviewer 2 Report

The authors took into account all the comments. The paper is a solid work and may be accepted

This manuscript is a resubmission of an earlier submission. The following is a list of the peer review reports and author responses from that submission.

Round 1

Reviewer 1 Report

In this paper, the authors present a new method to predict the detectability of peptides in mass spectrometry experiments, using machine learning. More specifically, they combine two existing neural network components, namely CapsNet and CBAM, to build a more sophisticated architecture. Then, they train their neural network on a large database of annotated peptides. They compare their method to other approaches for peptide detectability and peptide characterization. Eventually, they claim to perform better than related approaches.

Although the topic covered in this article could be of interest to the readership of IJMS, publication might be premature at this point. Indeed, numerous aspects of the methodology (especially the experimental setup) and of the results presentation are sub-standard. In addition, the poor quality of the English significantly hinders the reader's understanding of the presented work.

My major concerns are the following:
A) Many aspects of the evaluation methodology seem to be flawed, or are not presented at all. More specifically:
1) The authors do not perform any kind of cross-validation. According to section 4.1, they have divided the 100,000 peptides they use into a single training and testing sets pair. As a consequence, when the authors compare different settings and/or different methods, it is impossible to determine whether differences in performance are significant.
2) In section 2.1, the authors mention having repeated the experiment 10 times. First, it is unclear which experiment they are talking about. Second, how can they repeat the experiment, since they have only one training set and one testing set?
3) In section 2.1, what do the authors mean by testing different sets of features (namely separating or combining the biological features and the sequence features)? From what is explained in the methods section, these features are always separated into distinct inputs.
4) In section 2.2, what is the prediction task whose results are reported in table 1 and figure 2.a-c?
5) In section 2.3, what is the prediction task whose results are reported in table 2 and figure 2.d?
6) In section 2.4, what is the prediction task whose results are reported in table 3?
7) In section 2.5, how can the peptides for which predictions are made not be related to the training or testing sets, since they originate from the same database (namely GPMDB)?
8) In section 2.3, the frameworks against which the authors compare their work seem to be mostly irrelevant in this context.
9) At lines 257-258, the authors mention dividing 100 physicochemical properties into 10 groups of 10. Why not simply rewriting equations (2) and (3) so that they directly work on 100 physicochemical properties?

B) The authors seem to overstate many of their results. More specifically:
1) In section 2.2, after comparing several sets of input features, the authors state that "each feature has an important contribution in the detectability of the peptide". This doesn't seem to be true, as performance with or without amino acid frequency is very similar. In table 1, only sensitivity is a little lower without amino acid frequency. In Figure 2.b both ROC curves are almost identical.
2) In section 2.3, the authors state that their "model outperforms other frameworks in main evaluation indicators". However, table 2 shows that ResNet performs almost as well as their method. Without cross-validation, it is impossible to determine whether these differences are significant or not.
3) In section 2.5, because the prediction task is not clearly stated, it is not really possible to determine whether Capsnet is doing better than 1D-2C-CNN. Indeed, intuitively, one would expect peptides belonging to the "current missing proteins" (current MPs) set to not be detectable. Therefore, one could say that 1D-2C-CNN does a better job at differentiating these peptides from the detectable ones (in PE1 and Detected MPs).

I would also like to raise the following minors concerns:
C) Some parts of the manuscript are not clearly explained, or are sloppily explained. For example:
1) In several parts of the manuscript, the authors mention the "conical coordinate features", sometimes called "conic coordinates" or "conic residue coordinate features", without really explaining what they are talking about. As this is a pretty obscure concept, the bare minimum would be to add a reference, for the reader to understand that they have borrowed it from other work.
2) At line 294, the authors need a reference for the one-hot encoding.

D) The related work section is sub-standard, as the authors mostly cite old papers and/or unrelated work. For example:
1) At lines 38-39, the authors write that "Recently, researchers have proposed various methods to evaluate the detectability of peptides", but the articles they reference are from 2006 and 2010.
2) Work on cell-penetrating peptides and type prediction for peptides might not be the most closely related work. Surely, there are more articles on detectability prediction that would be more relevant here.

Author Response

We deeply appreciate the reviewer for the constructive comments which help us improve the quality of this paper again. We carefully revised the paper according to these comments. The revisions made to the paper are highlighted in blue in the new version, and we address the comments in the following response file point by point.

Reviewer 2 Report

In this study, Yu et al. proposed a deep learning framework based on CapsNet to predict peptide detectability. The authors claimed a better performance than previous studies on a benchmark dataset. However, there are some concerns that should be addressed:

1. English language should be improved.

2. For such kind of study, the authors should provide a web server to support testing their models as well as for practical use.

3. The authors did not show clearly the evaluation method that they used. For example, did they apply cross-validation on the results of ROC curve (Figure 2) or other metrics (tables)?

4. Statistical tests should be performed to see significant differences among different methods/models.

5. "Discussion" section lacks a lot of information. For example, the authors did not discuss or refer to other published studies.

6. Source codes should be released for replicating the methods.

7. Measurement metrics (i.e., sens, spec, acc, ...) have been used in previous bioinformatics-based studies such as PMID: 31920706 and PMID: 33260643. Thus, the authors are suggested to refer to more works in this description to attract a broader readership.

8. Quality of figures should be improved significantly.

9. In Figure 2, the authors should combine Fig. A, B, C, into a figure.

10. The authors should provide more detailed information on ROC curve title. Currently, all titles are "ROC" and the readers will be confused among figures.

11. More detail on hyperparameter tuning of all models should be listed.

Author Response

(The authors gave the same response as above.)

Round 2

Reviewer 1 Report

Unfortunately, it seems that the authors have only performed superficial changes in their manuscript. They have not really addressed the major shortcomings in their methodology.
My main concern, which was also shared by the other reviewer, is that the authors do not provide any way to judge whether the differences in performance between different settings/methods are significant. All they have done is add bar plots of F-scores. Since their performance results were obtained through 10-fold cross-validation, in addition to reporting averages across the 10 runs, the authors should at least report standard errors (especially in tables 2 and 3). Even better, as suggested by the other reviewer, the authors could provide statistical tests of the significance of differences in results. As it is, we still don't know whether we can trust the results that are reported in this study.
In their comparison to other neural network frameworks, the authors are still vastly over-stating their achievements. First, they compare their approaches to frameworks that were developed for totally different tasks ("TextCNN model for PPI site prediction, and the ResNet framework for predicting immunogenic peptide recognized by T cell receptor"). Despite being developed for a completely different purpose, ResNet performs almost as well as the authors' approach, putting their achievements in doubt.
Finally, numerous parts of the manuscript are still totally unclear. In addition to the English quality being very poor, some parts of the text are just not properly explained. It feels like the authors have tried to be as concise as possible, sometimes just using a few words to try and convey complex concepts. As far as I know, IJMS is not imposing any page limit. Therefore, the authors should not feel constrained by space limits, and they should clearly explain their methodology. The authors should also expand on their claims; they cannot just write sweeping statements, such as "These results show that [...] our model [...] is better than LSTM and TextCNN methods on main evaluation indicators." Each statement has to be backed up with data, and each claim should be clearly justified. As it stands most claims in this manuscript are only down to the subjective interpretation of the authors.

Reviewer 2 Report

My previous comments have been addressed well.